# Spatial radionuclide deposition data from the 60 km radial area around the Chernobyl nuclear power plant: results from a sampling survey in 1987

Valery Kashparov[1,3], Sviatoslav Levchuk[1], Marina Zhurba[1], Valentyn Protsak[1], Nicholas A. Beresford[2], and Jacqueline S. Chaplow[2]

[1] Ukrainian Institute of Agricultural Radiology of National University of Life and Environmental Sciences of Ukraine, Mashinobudivnykiv str.7, Chabany, Kyiv region, 08162 Ukraine
[2] UK Centre for Ecology & Hydrology, Lancaster Environment Centre, Library Avenue, Bailrigg, Lancaster, LA1 4AP, UK
[3] CERAD CoE Environmental Radioactivity/Department of Environmental Sciences,  Norwegian University of Life Sciences, 1432 Aas, Norway

*Correspondence to:* Jacqueline S. Chaplow (jgar@ceh.ac.uk)

**Abstract**. The data set "Spatial radionuclide deposition data from the 60 radial km area around the Chernobyl nuclear power plant: results from a sampling survey in 1987" is the latest in a series of data to be published by the Environmental Information Data Centre (EIDC) describing samples collected and analysed following the Chernobyl nuclear power plant accident in 1986. The data result from a survey carried out by the Ukrainian Institute of Agricultural Radiology (UIAR) in April and May 1987 and include sample site information, dose rate, radionuclide (zirconium-95, niobium-95, ruthenium-106, caesium-134, caesium-137 and cerium-144) deposition, and exchangeable (determined following 1M $NH_4Ac$ extraction of soils) caesium-134 and 137.
The purpose of this paper is to describe the available data and methodology used for sample collection, sample preparation, and analysis. The data will be useful in the reconstruction of doses to human and wildlife populations, answering the current lack of scientific consensus on the effects of radiation on wildlife in the Chernobyl Exclusion zone and in evaluating future management options for Chernobyl impacted area of Ukraine and Belarus.

The data and supporting documentation are freely available from the Environmental Information Data Centre (EIDC) under the terms and conditions of the Open Government Licence (Kashparov et al., 2019 https://doi.org/10.5285/a408ac9d-763e-4f4c-ba72-73bc2d1f596d).

## 1 Background

The dynamics of the releases of radioactive substance from the number four reactor at the Chernobyl nuclear power plant (ChNPP) and meteorological conditions over the ten days following the accident on the 26th April 1986 resulted in a complex pattern of contamination over most of Europe (IAEA, 2006).

The neutron flux rise and a sharp increase in energy emission at the time of the accident resulted in heating of the nuclear fuel and leakage of fission products. Destruction of the fuel rods caused an increase in heat transfer to the surface of the superheated fuel particles and coolant, and release of radioactive substances into the atmosphere (Kashparov et al., 1996). According to the latest estimates (Kashparov et al., 2003; UNSCEAR, 2008) 100% of inert radioactive gases (largely $^{85}Kr$ and $^{133}Xe$), 20-60% of iodine isotopes, 12-40% of $^{134,137}Cs$ and 1.4-4% of less volatile radionuclides ($^{95}Zr$, $^{99}Mo$, $^{89,90}Sr$, $^{103,106}Ru$, $^{141,144}Ce$, $^{154,155}Eu$, $^{238-241}Pu$ etc.) in the reactor at the moment of the accident were released to the atmosphere.

As a result of the initial explosion on 26th April 1986, a narrow (100 km long and up to 1 km wide) relatively straight trace of radioactive fallout formed to the west of the reactor in the direction of Red Forest and Tolsty Les village (this has subsequently become known as the

'western trace'). This trace was mainly finely dispersed nuclear fuel (Kashparov et al., 2003,
2018) and could only have been formed as a consequence of the short-term release of fuel
particles with overheated vapour to a comparatively low height during night time (the accident
occurred at 01:24) stable atmospheric conditions. At the time of the accident, surface winds
were weak and did not have any particular direction; only at a height of 1500 m was there a
south-western wind with the velocity 8-10 m·s$^{-1}$ (IAEA, 1992). Cooling of the release cloud,
which included steam, resulted in the decrease of its volume, water condensation and wet
deposition of radionuclides as mist (as the released steam cooled) (Saji, 2005). Later, the main
mechanism of fuel particle formation was the oxidation of the nuclear fuel (Kashparov et al.,
1996; Salbu et al., 1994). There was an absence of data on meteorological conditions in the
area of ChNPP at the time of the accident (the closest observations were more than 100 km
away to the west (Izrael et al., 1990)). There was also a lack of source term information and
data on the composition of dispersed radioactive fallout. Consequently, it was not possible to
make accurate predictions of deposition for the area close to the ChNPP (Talerko, 2005).
The relative leakage of fission products of uranium (IV) oxide in an inert environment at
temperatures up to 2600 °C decreases in the order: volatile (Xe, Kr, I, Cs, Te, Sb, Ag), semi-
volatile (Mo, Ba, Rh, Pd, Tc) and nonvolatile (Sr, Y, Nb, Ru, La, Ce, Eu) (Kashparov et al.,
1996; Pontillon et al., 2010). As a result of the estimated potential remaining heat release from
fuel at the time of the accident (~230 W kg$^{-1}$ U) and the heat accumulation in fuel (National
Report of Ukraine, 2011), highly mobile volatile fission products (Kr, Xe, iodine, tellurium,
caesium) were released from the fuel of the reactor and raised to a height of more than 1 km
on 26[th] April 1986 and to approximately 600 m over the following days (IAEA, 1992; Izrael et
al., 1990). The greatest release of radiocaesium occurred during the period of maximum heating
of the reactor fuel on 26-28[th] April 1986 (Izrael et al., 1990). This caused the formation of the
western, south-western (towards the settlements of Poliske and Bober), north-western
(ultimately spreading to Sweden and wider areas of western Europe), and north-eastern
condensed radioactive traces. Caesium deposition at distances from Chernobyl was largely
determined by the degree of precipitation (e.g. see Chaplow et al. (2015) discussing deposition
across Great Britain). After the covering of the reactor by dropping materials (including 40 t
of boron carbide, 2500 t of lead, 1800 t of sand and clay, 800 t of dolomite) from helicopters
over the period 27[th] April–10[th] May 1986 (National Report of Ukraine, 2011), the ability for
heat exchange of the fuel reduced, which caused a rise of temperature and consequent increase
of the leakage of volatile fission products and the melting of the materials which had been
dropped onto the reactor. Subsequently, there was a sharp reduction in the releases of
radionuclides from the destroyed reactor on 6[th] May 1986 (National Report of Ukraine, 2011)
due to aluminosilicates forming thermally stable compounds with many fission products and
fixing caesium and strontium at high temperature (a process known prior to the Chernobyl
accident (Hilpert & Nurberg, 1983)).
The changes of the annealing temperature of the nuclear fuel during the accident had a strong
effect on both the ratio of different volatile fission products released (the migratory properties
of Xe, Kr, I, Te, Cs increased with the temperature rise and were influenced by the presence of
$UO_2$) and the rate of destruction of the nuclear fuel which oxidised forming micronized fuel
particles (Salbu et al., 1994; Kashparov et al., 1996). The deposition of radionuclides such as
$^{90}$Sr, $^{238-241}$Pu, $^{241}$Am, which were associated with the fuel component of the Chernobyl releases
was largely limited to areas relatively close to the ChNPP. Areas receiving deposition of these
radionuclides were the Chernobyl Exclusion Zone (i.e. the area of approximately 30 km radius
around the ChNPP), and adjacent territories in the north of the Kiev region, in the west of the
Chernihiv region, and the Bragin and Hoyniki districts of the Gomel region (Belarus).
Deposition was related to the rate of the dry gravitational sedimentation of the fuel particles
caused by their high density (about 8-10 g·cm$^{-3}$ (Kashparov et al., 1996)); sedimentation of the
lightweight condensation particles, containing iodine and caesium radioisotopes, was lower
and hence these were transported further.
After the Chernobyl accident, western Europe and the Ukrainian-Belorussian Polessye were
contaminated with radionuclides (IAEA, 1991, 1992, 2006). However, the area extending to
60-km around the ChNPP was the most contaminated (Izrael et al., 1990). Work on the
assessment of the radiological situation within the zone started within a few days of the
accident; the aim of this work was the radiation protection of the population and personnel.
Subsequently, further quantification of terrestrial dose rates was carried out by aerial-gamma
survey by the State Hydrometeorological Committee together with Ministry of Geology and
Ministry of Defence of USSR (as reported in Izrael et al., 1990). Large-scale sampling of soil
was also conducted, with samples analysed using gamma-spectrometry and radiochemistry
methods (see Izrael et al., 1990). These studies showed high variability in dose rates and
radionuclide activity concentrations, with spatial patterns in both radioactive contamination
and the radionuclide composition of fallout (Izrael et al., 1990).
The initial area from which the population was evacuated was based on an arbitrary decision
whereby a circle around the Chernobyl nuclear power plant with a radius of 30 km was defined
(IAEA, 1991). In the initial phase after the accident (before 7$^{th}$ May 1986) 99195 people were
evacuated from 113 settlements including 11358 people from 51 villages in Belarus and 87 837
people from 62 settlements in Ukraine (including about 45 thousand people evacuated between
14.00-17.00 hours on April 27 from the town of Pripyat located 4 km from the ChNPP)
(Aleksakhin et al., 2001).
The analysis of data available in May 1986 showed that the extent of the territory with
radioactive contamination where comprehensive measures were required to protect the
population extended far beyond the 30 km Chernobyl Exclusion Zone (CEZ). A temporary
annual effective dose limit of 100 mSv for the period from 26$^{th}$ April 1986 to 25$^{th}$ April 1987
(50 mSv from external and 50 mSv from internal exposure) was set by the USSR Ministry of
Health. To identify areas outside of the CEZ where the population required evacuation, dose
criteria had to be defined. It was proposed to use the average value of the dose rate of gamma
radiation in open air for an area (estimated for 10$^{th}$ May 1986) to help define an evacuation
zone. An exposure dose rate of 5 mR h$^{-1}$ estimated for 10$^{th}$ May 1986 (approximating to an
effective dose rate (EDR) of gamma radiation in air of 50 µSv h$^{-1}$) equated to an external annual
dose of 50 mSv for the period from 26$^{th}$ April 1986 to 25$^{th}$ April 1987.
At the end of May 1986 an approach to identify areas where evacuation was required using
estimated internal dose rates was proposed. This used the average density of the surface
contamination of the soil with long-lived biologically significant nuclides ($^{137}$Cs, $^{90}$Sr, $^{239,240}$Pu)
in a settlement and modelling to estimate the contamination of foodstuffs and hence diet. The
numerical values suggested to identify areas for evacuation were: 15 Ci km$^{-2}$ (555 kBq m$^{-2}$) of
$^{137}$Cs, 3 Ci km$^{-2}$ (111 kBq m$^{-2}$) of $^{90}$Sr and 0.1 Ci km$^{-2}$ (3.7 kBq m$^{-2}$) of $^{239,240}$Pu; this equated
to an internal dose of 50 mSv over the first year after the accident.
However, in reality the main criterion for the evacuation was the exposure dose rate (R h$^{-1}$) and
where the exposure dose rate exceeded 5 mR h$^{-1}$ (EDR in air of about 50 μSv h$^{-1}$) the evacuated
population were not allowed to return.
Hence, in 1986 the boundary of the population evacuation zone was set at an exposure dose
rate of 5 mR h$^{-1}$ (EDR of about 50 μSv h$^{-1}$). However, the ratio of short-lived gamma-emitting
radionuclides ($^{95}$Zr, $^{95}$Nb, $^{106}$Ru, $^{144}$Ce) deposited as fuel particles to $^{134,137}$Cs deposited as
condensation particles, was inconsistent across the evacuated areas. Therefore, after the
radioactive decay of the short-lived radionuclides the residual dose rate across the evacuated
areas varied considerably and was largely determined by the pattern of long-lived $^{137}$Cs
deposition (e.g. Figure 1) (Kashparov et al., 2018).

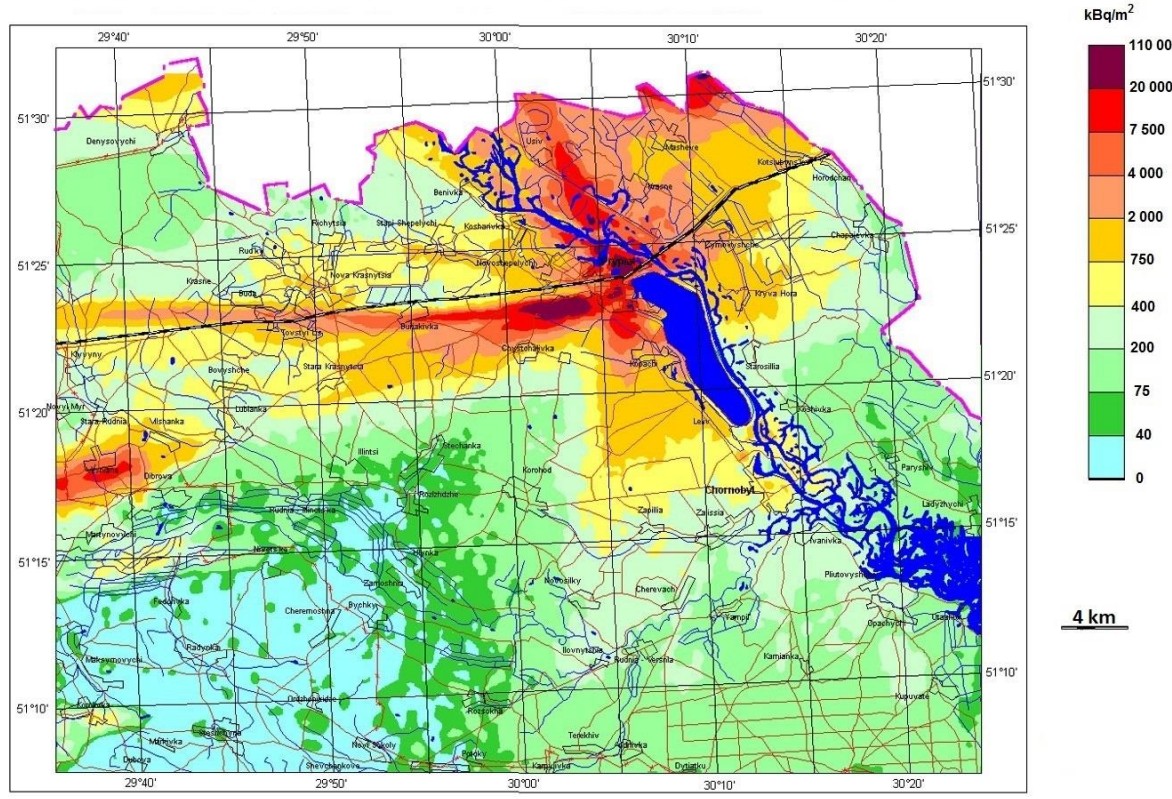


Figure 1. Caesium-137 deposition in the Ukrainian 30 km exclusion zone estimated for 1997
(from UIAR, 1998).

The first measurements of activity concentration of radionuclides in soil showed that
radionuclide activity concentration ratios depended on distance and direction from the ChNPP
(Izrael et al., 1990). Subsequent to this observation a detailed study of soil contamination was
started in 1987 (Izrael et al., 1990). Taking into account the considerable heterogeneity of
terrestrial contamination with radioactive substances in a large area, sampling along the
western, southern and northern traces was carried out in stages finishing in 1988.
In 1987 the State Committee of Hydrometeorology of the USSR and the Scientific Centre of
the Defence Ministry of the USSR established a survey programme to monitor radionuclide
activity concentrations in soil. For this purpose, 540 sampling sites were identified at a distance
of 5 km to 60 km around the ChNPP using a polar coordinate system centred on the ChNPP.
Fifteen sampling sites were selected on each of the 36 rays drawn every 10 degrees (Loshchilov

et al., 1991) (Figures 3 and 4). Radionuclide activity concentrations in soil samples collected on the radial network were determined by the UIAR and used to calculate the radionuclide contamination density. These data are discussed in this paper and the full data set is freely available from Kashparov et al. (2019).

## 2 Data

The data (Kashparov et al., 2019) include location of sample sites (angle and distance from the ChNPP), dose rate, radionuclide deposition data, counting efficiency and information on exchangeable $^{134,137}$Cs.

The data are presented in a table with 21 columns and 540 rows of data (plus column headings) as one Microsoft Excel Comma Separated Value File (.csv) as per the requirements of the Environmental Information Data Centre. Appendix 1 presents an explanation of the column headings and units used in the data (Kashparov et al., 2019).

### 2.1 Sampling

To enable long-term monitoring and contamination mapping of the 60 km zone around the ChNPP, 540 points were defined and sampled in April – May 1987. The sampling strategy used a radial network with points at every 10° (from 10° to 360°); sampling points were located at distances of 5 km, 6 km, 7 km, 8.3 km, 10 km, 12 km, 14.7 km, 17 km, 20 km, 25 km, 30 km, 37.5 km, 45 km, 52.5 km and 60 km (Figures 3 and 4). The locations of sampling points were identified using military maps (1:10000 scale) and local landscape. Sampling sites (identified using an index post) were estimated to be within 10 m of distances and directions as recorded in the accompanying data set. Sites were resampled regularly until 1990 and sporadically thereafter, however, data for these subsequent samplings are not reported here as they are unavailable (including to the UIAR).

Samples were not collected from points located in swamps, rivers and lakes; in total 489 samples were collected. A corer with a diameter of 14 cm was used to collect soil samples down to a depth of 5 cm from five points at each location using the envelope method (with approximately 5-10 m between sampling points) (Figure 2) (Loshchilov et al., 1991). Soil cores were retained intact during transportation to the laboratory. At each sampling point, the exposure dose rate was determined 1 m above ground level.

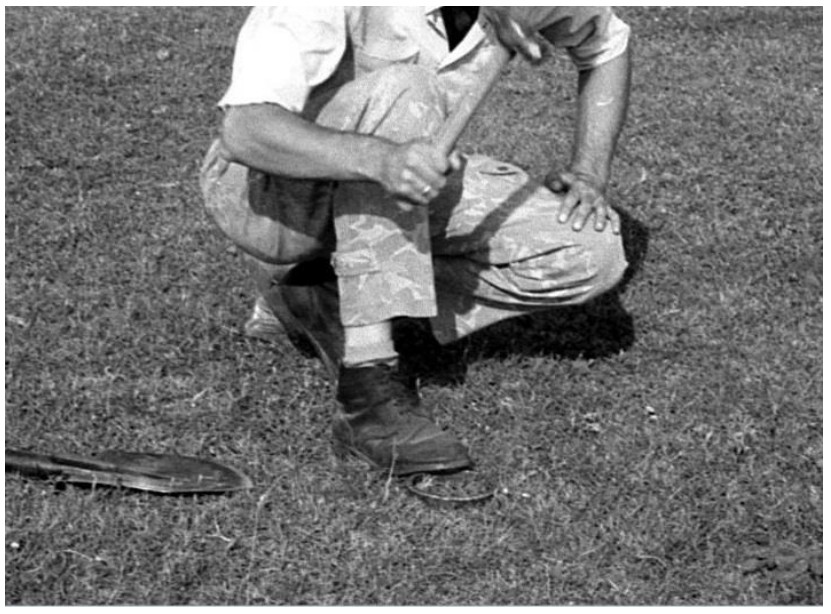

Figure 2. Soil sampling using a ring of 14 cm diameter to collect a 5 cm deep soil core (courtesy
of UIAR, 1989).

**2.2 Analysis**
Using a high-purity germanium detector (GEM-30185, ORTEC, USA) and a multichannel
analyser "ADCAM-300" (ORTEC, USA), the activity concentration of gamma emitting
radionuclides (zirconium-95 ($^{95}$Zr), niobium-95 ($^{95}$Nb), ruthenium-106 ($^{106}$Ru), caesium-134
($^{134}$Cs), caesium-137, ($^{137}$Cs) cerium-144 ($^{144}$Ce)) was determined in one soil sample from each
sampling site. Information on gamma lines used in the analyses and radioisotope half-lives
assumed for decay correction are presented in Appendix 2. Soil samples were analysed in a 1
litre Marinelli container. The other four cores were sent to different laboratories in the Soviet
Union (data for these cores are unfortunately not available). Using a 1M NH$_4$Ac solution (pH
7) a 100 g subsample of soil was leached (solid: liquid ratio 1:5). The resultant leachate solution
was shaken for 1 hour and then left at room temperature for 1 day before filtering through
ashless filter paper (3-5 µm). The filtrate was then put into a suitable container for gamma
analysis to determine the fraction of exchangeable $^{134,137}$Cs. Measured activity concentrations
were reported at 68% confidence level (which equates to one standard deviation).
Decay radiation information from the master library, integrated in spectrum analysing software
tool Gelicam (EG&G ORTEC, USA), was used in gamma-analyses. Activities of $^{106}$Ru and
$^{137}$Cs in samples were estimated via their gamma radiation emitting progenies $^{106}$Rh and $^{137m}$Ba,
respectively.

Calibration of the spectrometer was conducted using certified standards (soil equivalent multi-
radionuclide standard, V. G. Khlopin Radium Institute, Russia). Quality assurance/quality
control procedures included regular monitoring of the system performance, efficiency,
background and full width at half maximum (FWHM) for the $^{144}$Ce, $^{137}$Cs and $^{95}$Nb photo
peaks. To validate accuracy and precision of the method employed for $^{137}$Cs activity
concentration measurements, quality control samples (i.e., different matrix samples including
water, soil and sawdust spiked with known certified activities of radionuclides) and Certified
Reference Materials (CRM) were analysed alongside the samples. Analysis of IAEA CRMs
showed satisfactory results for radionuclide mean activity concentrations with results being
within the 95% confidence interval; the limit of detection for $^{137}$Cs in all samples was 1 Bq.
Subsamples were analysed in a different laboratory (USSR Ministry of Defence) and results
for the two laboratories were within the error of determination.

The density of soil contamination (Bq m$^{-2}$) was calculated from the estimated radionuclide
activity concentrations in soils. It has been estimated that uncertainty from using a single soil
sample (of area 0.015 m$^2$) to estimate the value of contamination density of a sampling site (i.e.
the area from which five cores were collected) may be up to 50% (IAEA, 2019).

The data described in this paper (Kashparov et al., 2020) comprise exposure dose rate (mR/h),
date of gamma activity measurement, density of contamination (Bq m$^{-2}$) of $^{95}$Zr, $^{95}$Nb, $^{106}$Ru,
$^{134}$Cs, $^{137}$Cs and $^{144}$Ce (with associated activity measurement uncertainties) and density of
contamination of $^{134+137}$Cs in exchangeable form. Reported radionuclide activity concentration
values are for the date of measurement (samples were analysed within 1.5 months of
collection).

For presentation below, radionuclide activity concentrations have been decay corrected to 6th
May 1986 (the date on which releases from the reactor in-effect stopped) using the equation:
$A_T = A_0/e^{-\lambda t}$ where $A_T$ equals the radionuclide activity concentration at the time of measurement
($t$); $A_O$ is the activity concentration on 6$^{th}$ May 1986, and $\lambda$ is the decay constant (i.e.
0.693/radionuclide physical half-life (see Table 1 for radionuclide half-lives)).

**2.3 Results**
The contamination density of $^{144}$Ce and $^{137}$Cs are presented in Figure 3 and 4; the activity
concentrations as presented in the figures have been decay corrected to 6th May 1986. The
density of $^{144}$Ce contamination decreased exponentially with distance (Figures 3 and 5),
because $^{144}$Ce was released in the fuel particles, which had a high dry deposition velocity
(Kuriny et al., 1993). The fallout density of $^{144}$Ce decreased by 7-9 times between the 5 km and
30 km sampling sites, and by 70-120 times between the 5 km and 60 km sampling sites (Figure
257  5).


The fallout density of $^{137}$Cs decreased similarly to that of $^{144}$Ce along the southern 'fuel trace'
(Figure 5a). The contamination density of $^{137}$Cs along the western trace decreased less than the
$^{144}$Ce contamination density due to the importance of the condensation component of the fallout
in this direction (with a resultant $R^2$ value for the relationship between $^{137}$Cs and distance lower
than seen for $^{144}$Ce and $^{137}$Cs in different directions) (Figure 5b). The comparative decrease of
$^{137}$Cs contamination density along the northern trace (mixed fuel and condensation fallout) was
in between that of the southern and western traces (Figure 5c) although there were caesium
hotspots in the northern condensation trace (Figures 4 and 5c). The activity ratio of $^{144}$Ce to
$^{137}$Cs decreased with distance from the ChNPP due to the condensation component being more
important for $^{137}$Cs; the condensation component had a lower deposition velocity compared
with fuel particles (with which $^{144}$Ce was associated) (Figure 6). The ratio $^{144}$Ce/$^{137}$Cs for
Chernobyl reactor fuel on 6$^{th}$ May 1986 can be estimated to be 15 from data presented in Table
1. The ratio was about 11 (geometric mean of 1167 measurements) in Chernobyl fuel particles
larger than 10 µm due to caesium escape during high-temperature annealing (Kuriny et al.,
1993). The ratio of $^{144}$Ce/$^{137}$Cs in deposition exceeded five in the south-east and in the south
up to 60 km and 30 km from the NPP respectively (Figure 6). Thus, activities of $^{134,137}$Cs in the
condensate and in the fuel components in these directions were of approximate equal
importance. The condensation component of caesium was more important in the north and
dominated in the west (Figure 8) (Loshchilov et al., 1991; Kuriny et al., 1993); the more rapidly
changing $^{144}$Ce/$^{137}$Cs ratios in these directions are reflective of this (Figure 6).

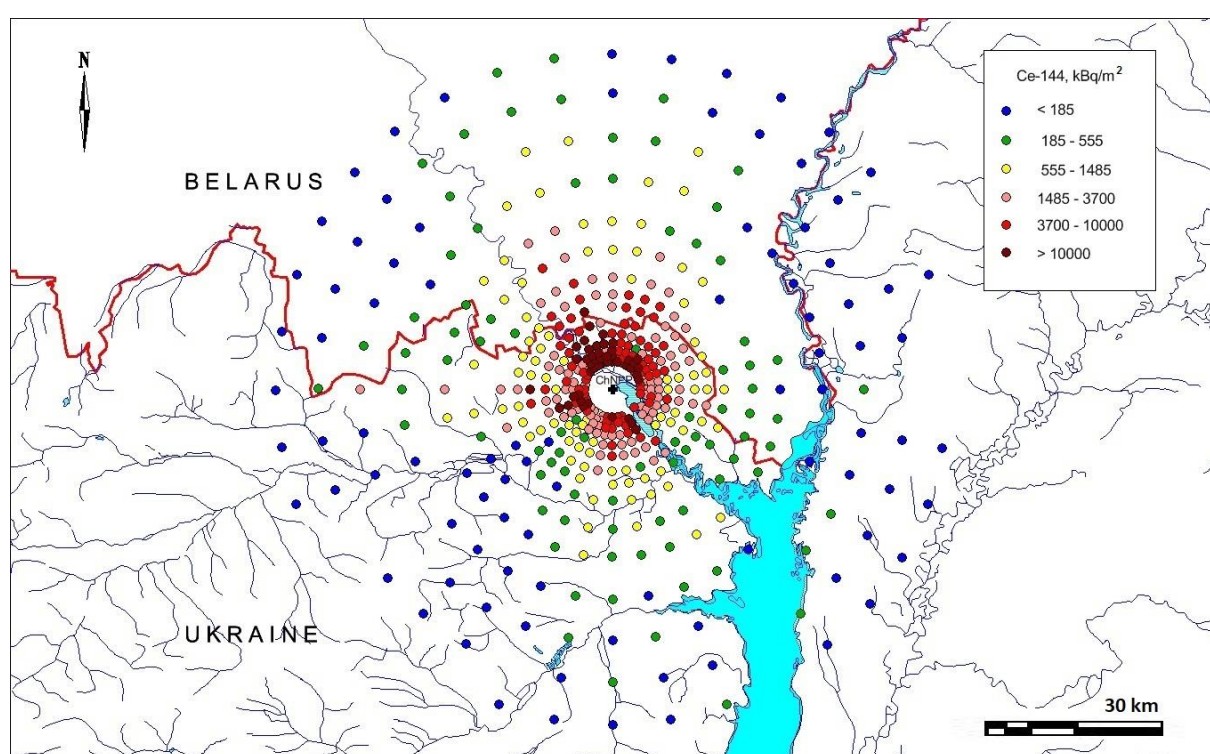


Figure 3. The fallout density of $^{144}$Ce (kBq/m$^2$) within the 60 km zone around the ChNPP
decay corrected to 6$^{th}$ May 1986.

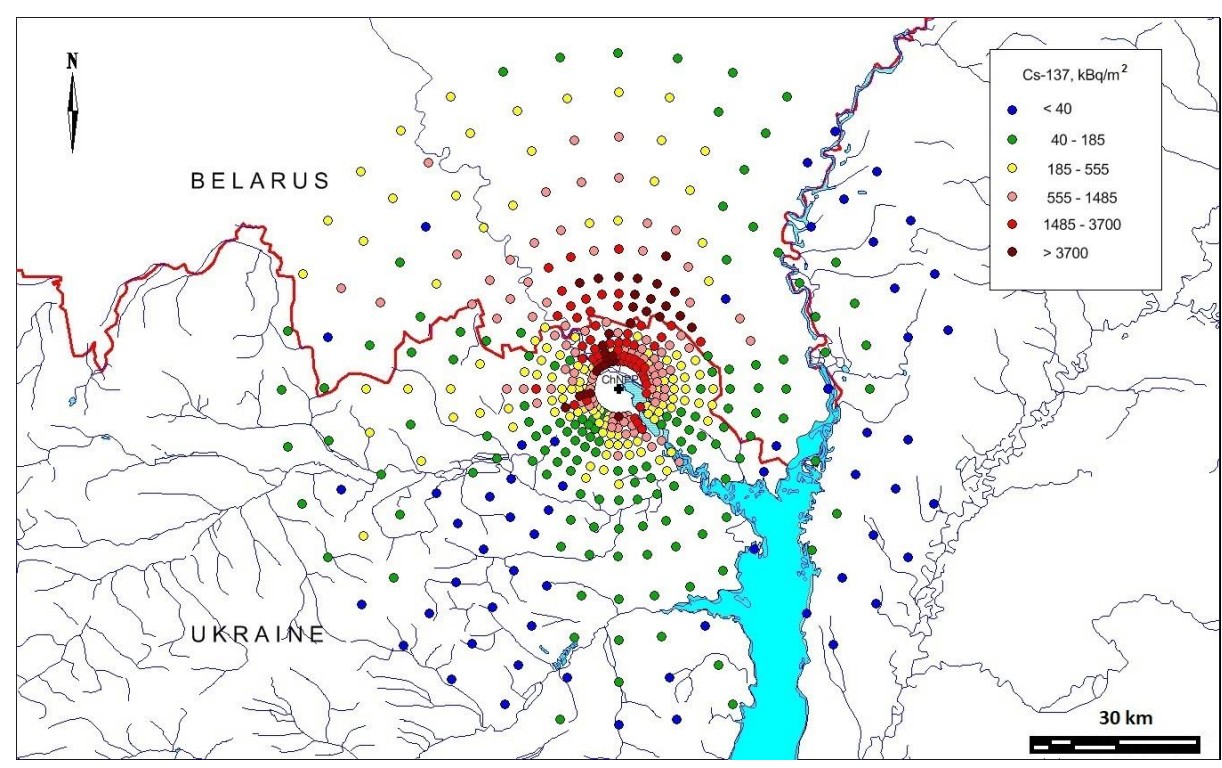


Figure 4. The fallout density of $^{137}$Cs (kBq/m$^2$) within the 60 km zone around the ChNPP
decay corrected to 6$^{th}$ May 1986.

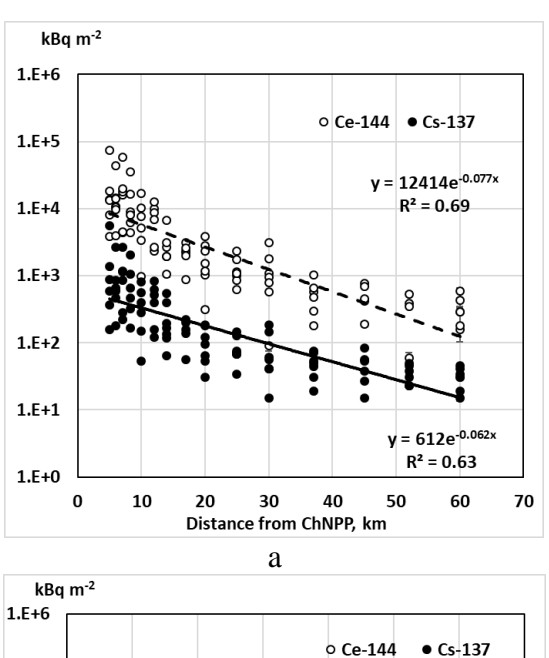

a


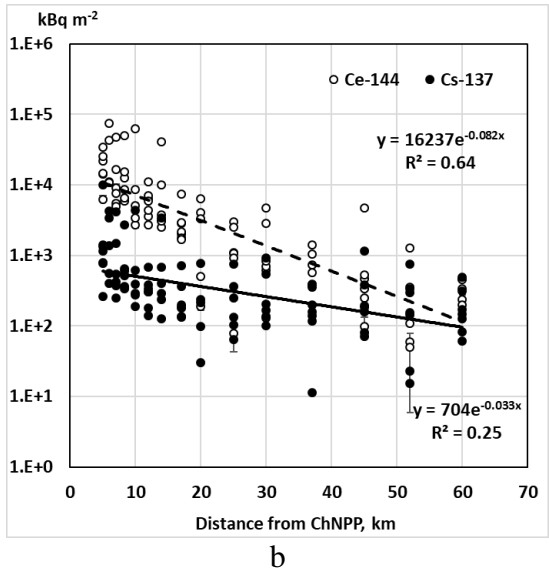

b


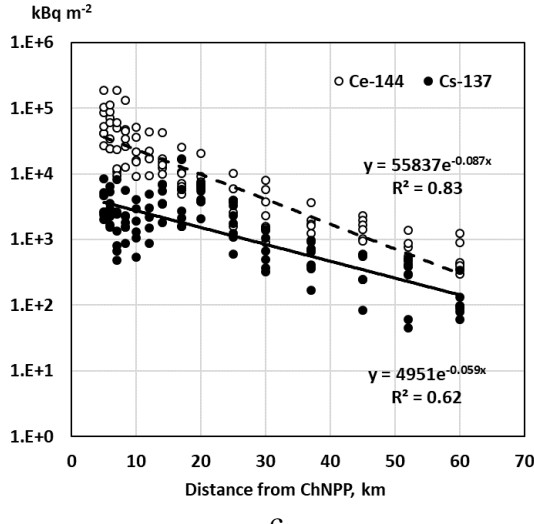

c

Figure 5. Relationship between fallout density of $^{144}$Ce (1) and $^{137}$Cs (2) and distance from
the ChNPP towards the south (a) (150-210°), the west (b) (240-300°) and the north (c) (330-
30°).

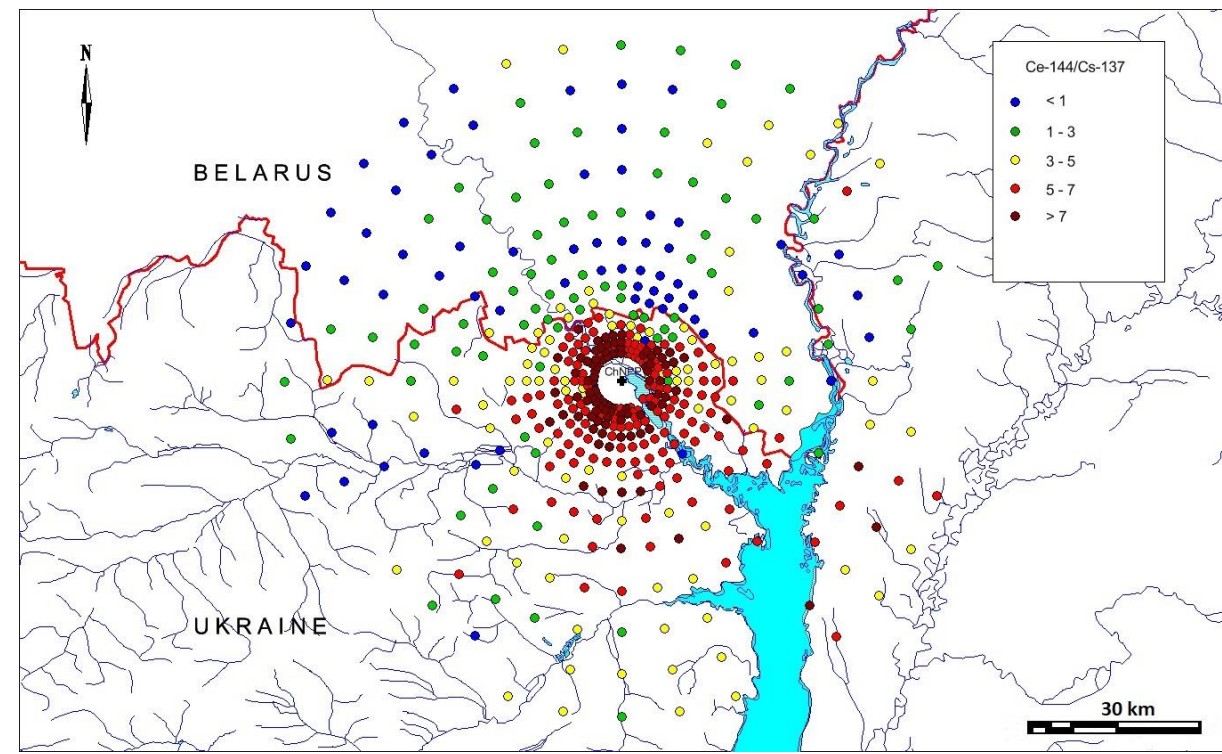

Figure 6. $^{144}$Ce/$^{137}$Cs ratio within the 60 km zone around the ChNPP decay corrected to 6$^{th}$
May 1986.

Table 1. The average activity concentrations of radionuclides with half-life (T$_{1/2}$) >1 day
estimated in the fuel of the ChNPP number four reactor recalculated for 6$^{th}$ May 1986
(Begichev et al., 1993).

| Radion uclide | Half-life (days) | Average activity concentration (Bq g$^{-1}$) | Radionuclide | Half-life (days) | Average activity concentration (Bq g$^{-1}$) |
|---|---|---|---|---|---|
| $^{75}$Se | 1.2 x 10$^2$ | 5.4 x 10$^6$ | $^{132}$Te | 3.3 x 10$^0$ | 2.4 x 10$^{10}$ |
| $^{76}$As | 1.1 x 10$^0$ | 1.7 x 10$^7$ | $^{133}$Xe | 5.2 x 10$^0$ | 3.4 x 10$^{10}$ |
| $^{77}$As | 1.6 x 10$^0$ | 4.1 x 10$^7$ | $^{134}$Cs | 7.6 x 10$^2$ | 8.9 x 10$^8$ |
| $^{82}$Br | 1.5 x 10$^0$ | 1.8 x 10$^9$ | $^{135}$Cs | 5.5 x 10$^7$ | 1.9 x 10$^4$ |
| $^{85}$Kr | 3.9 x 10$^3$ | 1.5 x 10$^8$ | $^{136}$Cs | 1.3 x 10$^1$ | 3.3 x 10$^{10}$ |
| $^{86}$Rb | 1.9 x 10$^1$ | 8.7 x 10$^9$ | $^{137}$Cs | 1.1 x 10$^4$ | 1.4 x 10$^9$ |
| $^{89}$Sr | 5.1 x 10$^1$ | 2.1 x 10$^{10}$ | $^{140}$Ba | 1.3 x 10$^1$ | 3.2 x 10$^{10}$ |
| $^{90}$Sr | 1.1 x 10$^4$ | 1.2 x 10$^9$ | $^{141}$Ce | 3.3 x 10$^1$ | 2.9 x 10$^{10}$ |
| $^{90}$Y | 1.1 x 10$^4$ | 1.2 x 10$^9$ | $^{143}$Ce | 1.4 x 10$^0$ | 2.9 x 10$^{10}$ |
| $^{91}$Y | 5.9 x 10$^1$ | 2.6 x 10$^{10}$ | $^{144}$Ce | 2.8 x 10$^2$ | 2.1 x 10$^{10}$ |
| $^{95}$Zr | 6.4 x 10$^1$ | 3.1 x 10$^{10}$ | $^{147}$Nd | 1.1 x 10$^1$ | 1.1 x 10$^{10}$ |
| $^{95}$Nb | 3.5 x 10$^1$ | 3.0 x 10$^{10}$ | $^{147}$Pm | 9.5 x 10$^2$ | 4.2 x 10$^9$ |

| | | | | | |
|---|---|---|---|---|---|
| $^{96}$Nb | $9.8 \times 10^1$ | $3.1 \times 10^{10}$ | $^{148m}$Pm | $4.1 \times 10^1$ | $8.5 \times 10^9$ |
| $^{99}$Mo | $2.7 \times 10^0$ | $3.2 \times 10^{10}$ | $^{149}$Nd | $2.2 \times 10^0$ | $5.8 \times 10^9$ |
| $^{99m}$Tc | $2.7 \times 10^0$ | $2.8 \times 10^{10}$ | $^{151}$Pm | $1.2 \times 10^0$ | $2.6 \times 10^9$ |
| $^{103}$Ru | $3.9 \times 10^1$ | $2.0 \times 10^{10}$ | $^{151}$Sm | $3.3 \times 10^4$ | $3.4 \times 10^7$ |
| $^{105}$Rh | $1.5 \times 10^0$ | $1.0 \times 10^{10}$ | $^{153}$Sm | $1.9 \times 10^0$ | $1.1 \times 10^9$ |
| $^{106}$Ru | $3.7 \times 10^2$ | $4.5 \times 10^9$ | $^{154}$Eu | $3.1 \times 10^3$ | $3.7 \times 10^7$ |
| $^{110m}$Ag | $2.5 \times 10^2$ | $5.3 \times 10^8$ | $^{155}$Eu | $1.7 \times 10^3$ | $4.85 \times 10^7$ |
| $^{111}$Ag | $7.5 \times 10^0$ | $4.4 \times 10^8$ | $^{156}$Eu | $1.5 \times 10^1$ | $1.9 \times 10^8$ |
| $^{115m}$In | $1.9 \times 10^1$ | $8.6 \times 10^7$ | $^{160}$Tb | $7.2 \times 10^1$ | $1.0 \times 10^7$ |
| $^{117m}$Sn | $1.4 \times 10^1$ | $8.3 \times 10^7$ | $^{237}$Np | $7.8 \times 10^8$ | $1.4 \times 10^3$ |
| $^{123}$Sn | $1.3 \times 10^2$ | $9.9 \times 10^7$ | $^{239}$Np | $2.4 \times 10^0$ | $3.1 \times 10^{11}$ |
| $^{124}$I | $4.2 \times 10^0$ | $1.4 \times 10^8$ | $^{236}$Pu | $1.0 \times 10^3$ | $6.0 \times 10^2$ |
| $^{125}$Sb | $1.0 \times 10^3$ | $7.8 \times 10^7$ | $^{238}$Pu | $3.2 \times 10^4$ | $6.8 \times 10^6$ |
| $^{125m}$Te | $5.8 \times 10^1$ | $1.6 \times 10^7$ | $^{239}$Pu | $8.8 \times 10^6$ | $5.0 \times 10^6$ |
| $^{126m}$Sb | $1.2 \times 10^1$ | $4.4 \times 10^8$ | $^{240}$Pu | $2.4 \times 10^6$ | $7.8 \times 10^6$ |
| $^{126}$Sb | $1.2 \times 10^1$ | $6.1 \times 10^7$ | $^{241}$Pu | $5.1 \times 10^3$ | $9.6 \times 10^8$ |
| $^{127}$Sb | $3.8 \times 10^0$ | $1.1 \times 10^9$ | $^{242}$Pu | $1.4 \times 10^8$ | $1.5 \times 10^4$ |
| $^{127}$Te | $1.1 \times 10^2$ | $8.9 \times 10^8$ | $^{241}$Am | $1.6 \times 10^5$ | $8.7 \times 10^5$ |
| $^{129m}$Te | $3.3 \times 10^1$ | $5.5 \times 10^9$ | $^{243}$Am | $2.7 \times 10^6$ | $5.1 \times 10^4$ |
| $^{131}$I | $8.0 \times 10^0$ | $1.6 \times 10^{10}$ | $^{242}$Cm | $1.6 \times 10^2$ | $2.3 \times 10^8$ |
| $^{131m}$Xe | $1.2 \times 10^1$ | $1.8 \times 10^8$ | $^{244}$Cm | $6.6 \times 10^3$ | $2.2 \times 10^6$ |


A good correlation ($R^2$=0.98) was observed between fallout densities of $^{95}$Zr (estimated from the activity concentration of daughter product $^{95}$Nb)[1] and $^{144}$Ce (Figure 7a) because both radionuclides were released and deposited as fuel particles (Kuriny et al., 1993; Kashparov et al., 2003; Kashparov, 2003). The fallout density ratio of $^{144}$Ce/$^{95}$Zr=0.73±0.05, decay corrected to 6th May 1986 was similar to that estimated for Chernobyl reactor fuel ($^{144}$Ce/$^{95}$Zr=0.68) (Table 1).

The activity ratio of $^{144}$Ce to $^{106}$Ru in fallout was correlated ($R^2$=0.93) and was 3.9±0.4 decay corrected to 6$^{th}$ May 1986 (Figure 7b). The value was close to the ratio of $^{144}$Ce/$^{106}$Ru estimated for fuel in the ChNPP number four reactor (4.7) (Table 1). Excess $^{106}$Ru activity relative to $^{144}$Ce activity in some soil samples was observed likely due to the presence of "ruthenium particles" (a matrix of iron group elements with a high content of $^{103,106}$Ru (Kuriny et al., 1993; Kashparov et al., 1996)).

There was a weak correlation ($R^2$=0.41) between $^{144}$Ce and $^{137}$Cs activities in the fallout because, as already discussed, caesium was largely deposited as condensation particles while cerium was deposited in fuel particles only. However, in highly contaminated areas close to the ChNPP a significant part of the $^{137}$Cs was deposited as fuel particles and the activity ratio of $^{144}$Ce/$^{137}$Cs of 9.1 (Figure 7c) broadly corresponded to that of 15 in the reactor fuel (Table 1).

---

[1] Niobium-95 ($T_{1/2}$=34 days) is the daughter radionuclide of $^{95}$Zr ($T_{1/2}$=65 days) and the ratio of their activities at an equilibrium equals $^{95}$Nb/$^{95}$Zr=2.1.

324

Different radioisotopes of caesium escaped from nuclear fuel and were deposited in the same way. This similar behaviour of $^{134}$Cs and $^{137}$Cs resulted in a strong correlation (R$^2$=0.99) between their activities in soil samples and the ratio of $^{134}$Cs/$^{137}$Cs=0.57±0.07 was similar to that estimated for the reactor fuel (0.64, Table 1).

329

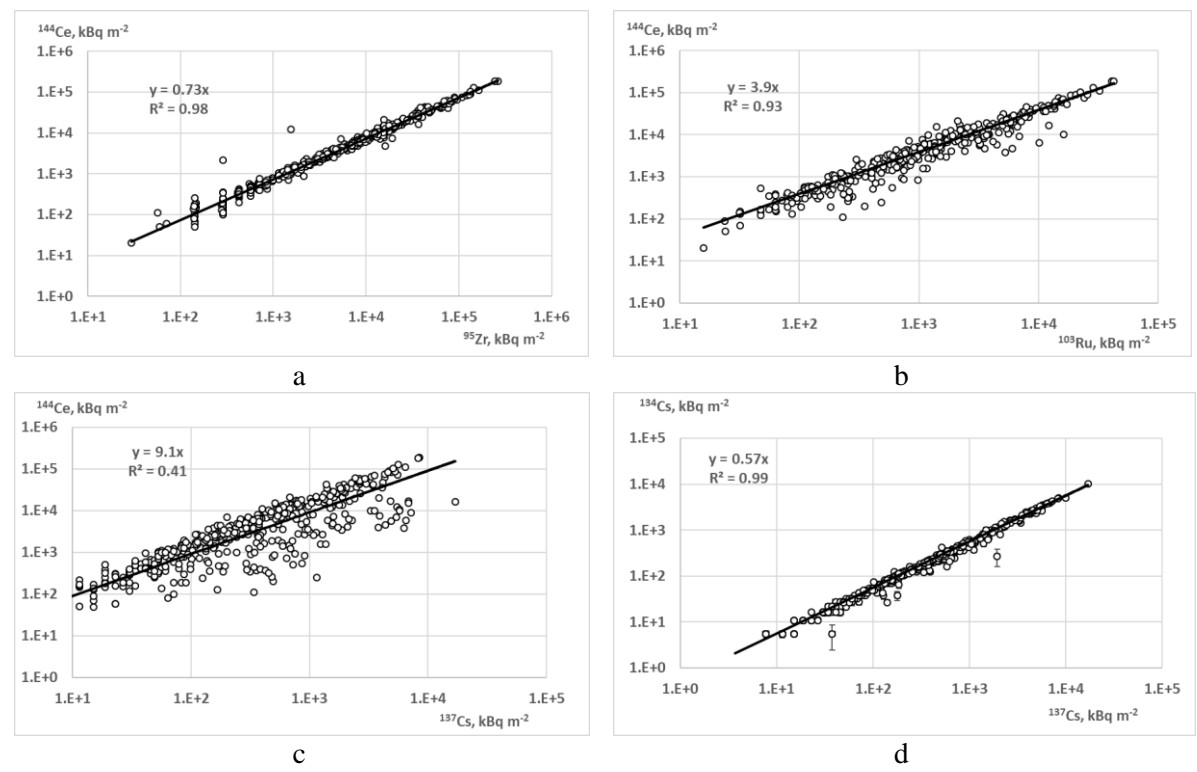

Figure 7. Correlation between deposition densities of different radionuclides decay corrected to 6$^{th}$ May 1986.

332

### 3 Use of the data

Apart from adding to the available data with which contamination maps for the CEZ and surrounding areas can be generated (e.g. Kashparov et al., 2018) the data discussed in this paper can be used to make predictions for less well studied radionuclides.

The determination of beta and alpha emitting radionuclides in samples requires radiochemical extraction which is both time consuming and relatively expensive. Large-scale surveys of the deposition of alpha and beta emitting radionuclides are therefore more difficult than those for gamma-emitting radionuclides and are not conducive with responding to a large-scale accident such as that which occurred at Chernobyl. Above we have demonstrated that the deposition behaviour of different groups of radionuclides was determined by the form in which they were present in the atmosphere (i.e. associated with fuel particles or condensation particles).

We propose that $^{144}$Ce deposition can be used as a marker of the deposition of fuel particles; fuel particles were the main deposition form of nonvolatile radionuclides (i.e. Sr, Y, Nb, Ru, La, Ce, Eu, Np, Pu, Am, Cm). Therefore, using $^{144}$Ce activity concentrations determined in soil samples and estimates of the activities in reactor fuel, we can make estimates of the deposition

of radionuclides such as Pu-isotopes and Cm that have been relatively less studied. For example, activity ratios of $^{238}$Pu, $^{239}$Pu $^{240}$Pu and $^{241}$Pu to $^{144}$Ce, at the time of measurement would be $8.4 \times 10^{-4}$, $6.2 \times 10^{-4}$, $9.7 \times 10^{-4}$ and $1.1 \times 10^{-1}$ respectively (estimated by decay correcting data presented in Table 1). Fallout densities of these plutonium isotopes can therefore be calculated for all sampling points where deposition density of $^{144}$Ce was measured either in this study (e.g. Figure 3) or in other data sets. As an example of the application of the data in this manner, Figure 8 presents the estimated deposition of $^{238}$Pu; Figure 8 was prepared using the TIN (triangulated irregular network) interpolation within MAPINFO. The first maps of $^{90}$Sr and $^{239+240}$Pu surface contamination from the Chernobyl accident were prepared in the frame of an international project (IAEA, 1992) in a similar way.

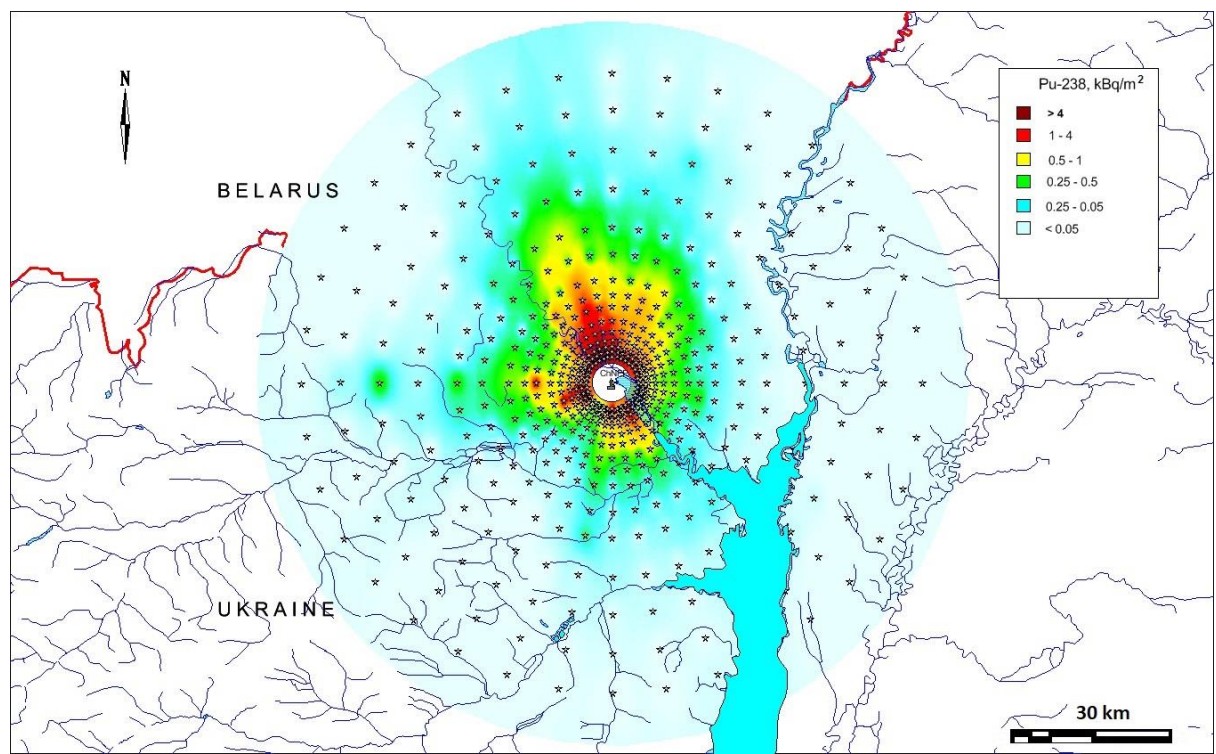

Figure 8. The fallout density of $^{238}$Pu (kBq m$^{-2}$) corrected to 6$^{th}$ May 1986; estimated from measurements of $^{144}$Ce in soil and estimated activity concentrations in the fuel of the ChNNP reactor number four (note no data were available for less than 5 km from ChNPP and no interpolation for this area has been attempted).

The dynamic spatial distribution of gamma dose rate can be reconstructed using the data on radionuclide contamination densities (Kashparov et al., 2019) in combination with the ratios between activities of radionuclides in fuel and in condensed components of Chernobyl fallout (Table 1) and also dose coefficients for exposure to contaminated ground surfaces, (Sv s$^{-1}$/Bq m$^{-2}$) (Eckerman & Ryman, 1993). Five days after deposition the following radionuclides were major contributors (about 95 %) to gamma dose rate: $^{136}$Cs, $^{140}$La, $^{239}$Np, $^{95}$Nb, $^{95}$Zr, $^{131}$I, $^{148m}$Pm, $^{103}$Ru, $^{140}$Ba, $^{132}$Te. After three months the major external dose contributors were: $^{95}$Nb, $^{95}$Zr, $^{148m}$Pm, $^{134}$Cs, $^{103}$Ru, $^{137m}$Ba, $^{110m}$Ag, $^{136}$Cs, $^{106}$Rh. Three years after the major contributors were $^{137m}$Ba, $^{134}$Cs, $^{106}$Rh, $^{110m}$Ag, $^{154}$Eu. At the present time the gamma dose can be estimated to be mainly (99%) due to the gamma-emitting daughter radionuclide of $^{137}$Cs ($^{137m}$Ba). Bondar (2015) from a survey of the CEZ along the Ukrainian-Belarussian border, showed a good relationship between $^{137}$Cs contamination ($A_{Cs-137}$, in the range of 17-7790 kBq m$^{-2}$) and

ambient dose rates at 1m above the ground ($D_{ext}$, in the range of 0.1-6.0 µSv h$^{-1}$).  The
relationship was described by following equation with correlation coefficient of 0.99:
$$D_{ext} = 0.0009 \cdot A_{Cs-137} + 0.14.$$
As an example of the application of the data in this manner, Figure 9 presents the estimated
external effective gamma dose rate five and 95 days after the cessation of the radioactive
releases from the reactor on 6$^{th}$ May 1986.

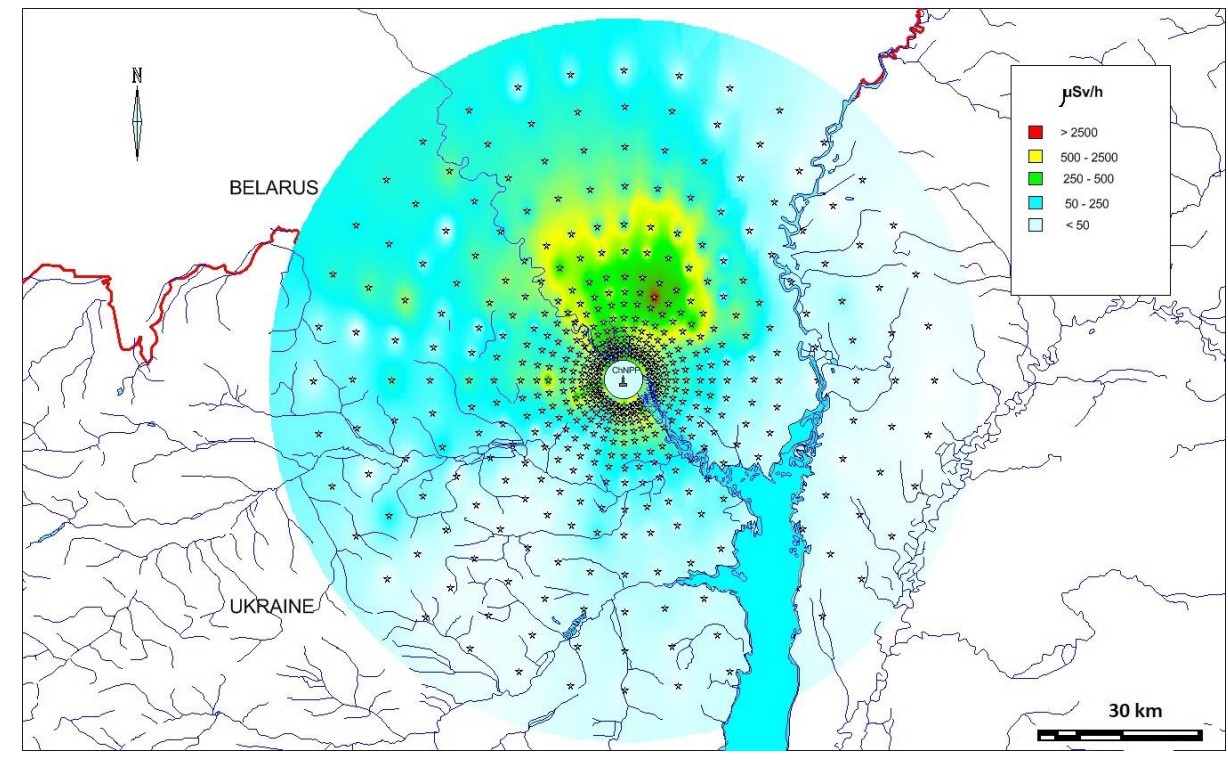

383                                                                              a


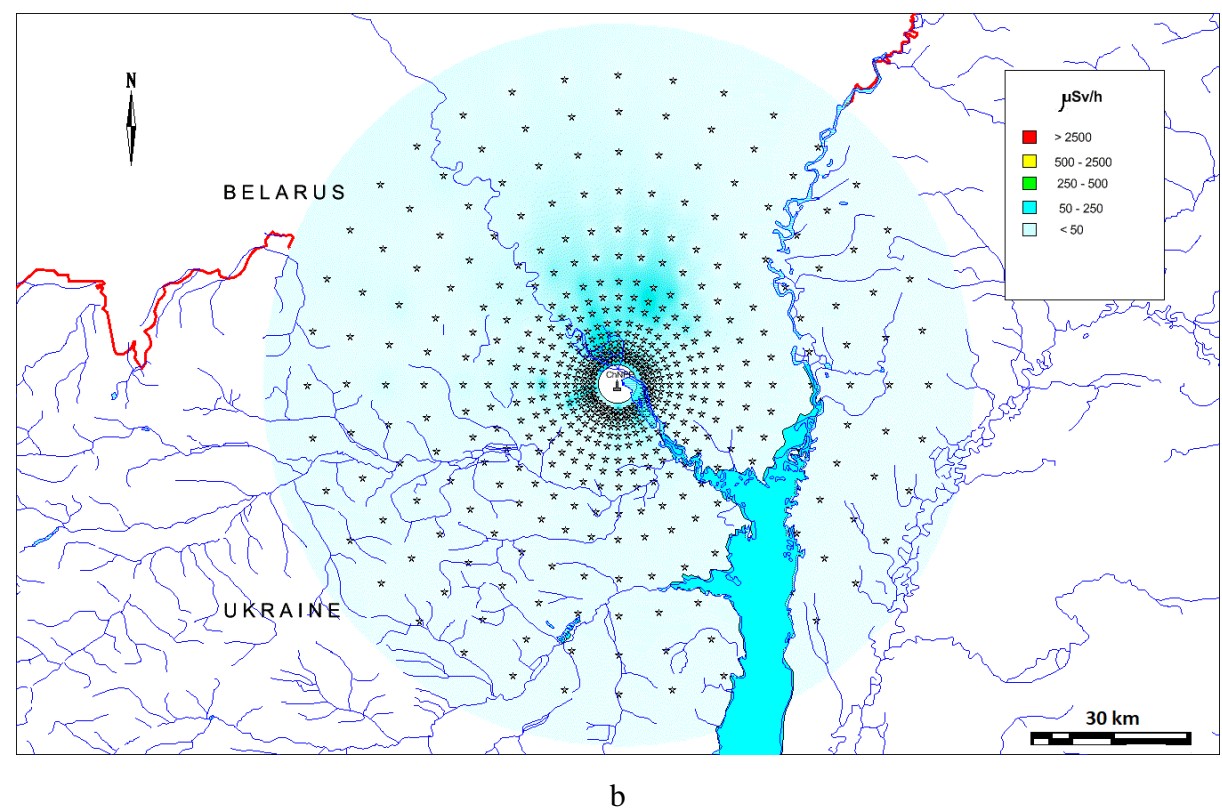


b

Figure 9. Spatial distribution, interpolated as for Figure 8, of effective dose rate within the 60 km zone around the ChNPP on $10^{th}$ May 1986 (a) and $10^{th}$ August 1986 (b). Note no data were available for less than 5 km from ChNPP and no interpolation for this area has been attempted.

390

The estimated effective dose rate values exceed the evacuation dose criteria of 50 μSv h$^{-1}$ over a large area (especially in the north and west) of the 60 km area around the ChNPP on $10^{th}$ May 1986 (Figure 9a); as discussed above a dose rate of 50 μSv h$^{-1}$ on $10^{th}$ May 1986 equated to a total dose over the first year after the accident of 50 mSv - the value used to define areas for evacuation. On the $10^{th}$ August 1986 the area estimated to exceed 50 μSv h$^{-1}$ was restricted to the north (Figure 9b). The dose rate decreased quickly after the accident due to the radioactive decay of short-lived radionuclides. The dominance of these short-lived radionuclides and a lack of knowledge of the radionuclide composition of the fallout made it difficult in 1986 to estimate external dose rates to the public for an evaluation date of $10^{th}$ May 1986 (most dose rate measurements being made after the $10^{th}$ May). This likely resulted in the overestimation of dose rates for some villages in 1986 leading to their evacuation when the external dose rate would not have been in excess of the 50 mSv limit used by the authorities.

There is a need for deposition data for the CEZ and surrounding areas for a number of reasons. These include exploring risks associated with future management options for the CEZ (e.g. management of the water table, forest fire prevention, increased tourism, etc.) and also the return of abandoned areas outside of the CEZ to productive use. The long-term effect of radiation exposure on wildlife in the CEZ is an issue of much debate (e.g. see discussion in Beresford et al., 2019). Improved data which can be used to map the contamination of a range of radionuclides will be useful in improving dose assessments to wildlife (including retrospective assessments of earlier exposure rates). The CEZ has been declared a

‘Radioecological Observatory’ (Muikku et al., 2018) (where a Radioecology Observatory is defined as a radioactively contaminated field site that provides a focus for joint, long-term, radioecological research). The open provision of data as described in this paper fosters the spirit of collaboration and openness required to make the observatory site concept successful and joins a growing amount of data made available for the CEZ (Kashparov et al., 2017; Fuller et al., 2018; Kendrick et al., 2018; Gaschak et al., 2018; Beresford et al., 2018; Lerebours and Smith, 2019).

## 4 Data availability

The data described here have a digital object identifier (doi: 10.5285/a408ac9d-763e-4f4c-ba72-73bc2d1f596d) and are freely available for registered users from the NERC Environmental Information Data Centre (http://eidc.ceh.ac.uk/) under the terms of the Open Government Licence (Kashparov et al., 2019).

Competing interests. The authors declare that they have no conflict of interest.

Acknowledgements. Funding for UKCEH staff to contribute to preparing this paper and the accompanying data set (Kashparov et al., 2019) was provided by the TREE project (http://www.ceh.ac.uk/tree; funded by NERC, the Environment Agency and Radioactive Waste Management Ltd under the RATE programme) and associated iCLEAR (https://tree.ceh.ac.uk/content/iclear-0; funded by NERC) projects.

Author contribution. Soil samples were collected by the USSR Ministry of Defence and delivered to UIAR. Sample preparation, analysis and data interpretation was carried out by UIAR staff contributing as follows: Kashparov, Levchuk, Protsak, - sample preparation, measurement of radionuclide activity concentrations in samples; Kashparov - analysis of results; Zhurba - database creation and preparation of the manuscript figures (maps). The manuscript was prepared by Chaplow, Beresford, Kashparov, Levchuk and Zhurba.

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

Appendix 1. A detailed explanation of the column headings and units (where applicable) which
accompanies the data (Kashparov et al., 2019).

| Column_heading | Explanation | Units |
|---|---|---|
| Identifier | Unique identification number | not applicable |
| Angle_degree | A number between 10 and 360 indicates the direction from the ChNPP in degrees; 90 degrees is due east, 180 degrees is due south, 270 degrees is due west and 0/360 degrees is due north. See Figure 1. | degree |
| Distance_from_ChNPP_km | Distance from the Chernobyl Nuclear Power Plant (ChNPP) reactor number 4 in kilometres | kilometres |
| Date_gamma_measurement | Date of gamma measurement. An empty cell indicates a network point located in a water body where sample collection was not possible | dd-month-yyyy |
| Exposure_dose_rate_mR/h | Dose rate in air at a height of 1 metre | milliroentgen per hour |
| Absorbed_dose_rate_microGray/h | Absorbed dose rate is the energy deposited in matter by ionizing radiation per unit mass | Micro Gray per hour |
| Zr-95_Bqm$^2$ | Density of soil contamination with zirconium-95 | Becquerel per square metre |
| Zr-95_relative_error | Relative uncertainty in determination of Zr-95 (at 68% confidence interval) | percentage |
| Nb-95_Bqm$^2$ | Density of soil contamination with niobium-95 | Becquerel per square metre |
| Nb-95_relative_error | Relative uncertainty in determination of Nb-95 (at 68% confidence interval) | percentage |
| Ru-106_Bqm$^2$ | Density of soil contamination with ruthenium-106 | Becquerel per square metre |
| Ru-106_relative_error | Relative uncertainty in determination of Ru-106 (at 68% confidence interval) | percentage |
| Cs-134_Bqm$^2$ | Density of soil contamination with caesium-134 | Becquerel per square metre |

| | | |
|---|---|---|
| Cs-134_relative_error | Relative uncertainty in determination of Cs-134 (at 68% confidence interval) | percentage |
| Cs-137_Bqm$^2$ | Density of soil contamination with caesium-137 | Becquerel per square metre |
| Cs-137_relative error | Relative uncertainty in determination of Cs-137 (at 68% confidence interval) | percentage |
| Ce-144_Bqm$^2$ | Density of soil contamination with cerium-144 | Becquerel per square metre |
| Ce-144_relative_error | Relative uncertainty in determination of Ce-144 (at 68% confidence interval) | percentage |
| Exch_Cs-134+Cs-137_Bqm$^2$ | Density of soil contamination with the exchangeable form of caesium | Becquerel per square metre |
| Note on empty cells | An empty cell means that data is not available | |
| Instrument | Gamma spectrometer with a semiconductor detector GEM-30185 ORTEC (results reported at 68% confidence level) | |


Appendix 2. Decay radiation information from the master library, integrated in spectrum
analysing software tool Gelicam (EG&G ORTEC, USA), used in gamma-analyses. Activities
of $^{106}$Ru and $^{137}$Cs in samples were estimated via their gamma radiation emitting progenies
$^{106}$Rh and $^{137m}$Ba, respectively

| Target radionuclide | Measured radionuclide | Energy, keV | Emission probability % | Half life of target radionuclides |
|---|---|---|---|---|
| $^{95}$Zr | 95Zr | 724.20 756.72 | 44.10 54.50 | 64.02 days |
| $^{95}$Nb | 95Nb | 765.79 | 99.79 | 34.97 days |
| $^{106}$Ru | 106Rh | 621.84 1050.47 | 9.812 1.73 | 368.2 days |
| $^{134}$Cs | 134Cs | 604.70 795.85 | 97.56 85.44 | 753.1 days |
| $^{137}$Cs | 137mBa | 661.66 | 85.21 | 30.174 years |
| $^{144}$Ce | 144Ce | 133.54 | 10.8 | 284.3 days |
