# Peer review of "Spatial radionuclide deposition data from the 60 km radial area around the"

_Earth System Science Data, 2019_

## Referee Comment (RC1) · Anonymous Referee #1 · 24 Feb 2020

General comments

This paper describes in detail the spatial radioactive contamination by condensation and fuel of the fallout caused by the Chernobyl accident in 1986, i.e. comparing 144Ce and 137Cs. Making these data available is, as nicely described in chapter 3 "Use of the data" important for assessing the long term effect of radiation exposure of the surrounding landscape including wildlife. The introduction is well written and interesting to read. As a geologist I was missing that today 137Cs deposited in 1986 is commonly used in areas far away from ChNPP to date sediment layers for environmental reconstructions

(e.g. doi: 10.1111/j.1365-3091.2012.01343.x.).

I favor Figure 9 because here you can see the development over time (May vs. August 1986), whereas other figures show only the static situation reconstructed for 6th May 1986.

When I looked into the data provided, I found a csv table with 20 parameters listed for 491 measurements between 15.05.1987 and 08.06.1987. 49 entries had ID's but no data. Metadata provide explanations and units as well as methods for the shown parameters. In the metadata it was described that missing values are due to water bodies. In the manuscript the authors state that the data include northing and easting, but I could not find coordinates in the data set. Is this missing by mistake?

Overall the study is presented in a good way. My concern is that the data are presented as "corrected to 6th May 1986", but obviously based on measurements roughly one year later in 1987. If the data are extrapolations back in time, the authors should describe in detail their methods how they calculated/corrected the values presented in the figures.

Specific comments:

Line 20-21 is this a redundant listing of "caesium-134 and caesium-137" or is there a striking difference? If so, maybe few words explaining why would help.

Line 22 You used exactly the same sentences as in the previous paper in ESSD. Please specify "them" in this context.

Line 35 Please provide a rough estimate of the vast area size.

Line 105-111 Describe how many samples and the spatial resolution of sampling (compare lines 159-160)

Line 168 I could not find Northing or Easting in the data set.

Line 178 More precise for "regularly" in which temporal resolution? Did sampling take

place at exactly the same locations? The photo shows that the upper column of the soil and grass was sampled. How did the resampling account for accumulation on top of the contaminated layer in subsequent years?

Line 198 – 200 – clarification on why data is not available – embargo or not processed?

Line 208 – 210 – uncertainty seems to be high - are there other means to check, whether the uncertainty could be limited? How did you calculate the 50%, is it standard deviation between 5 samples??

Figure 5b. – Why is the $R^2$ = 0.25 not discussed?

Line 230 – 232, which is associated with the plot, does not include specifics.

Technical comments:

Line 33 Is this the correct citation format (Chernobyl, 1996)?

Line 70 . . ."radiocaesium"?

Figure 1. It would help to remove blue color from legend, if it is not used, or use different color instead, because it is too close to the blue of the rivers and lakes. Is there a limit at the top of the legend?

Table 1. Scientific notation seems not very reader-friendly and the table seems long compared to the intended message. It would help if you could reduce it to a smaller number or highlight entries according to a meaningful criterion.

Line 181-182 repetitive statement to line 162?

---

## Referee Comment (RC2) · Anonymous Referee #2 · 26 Feb 2020

General comments

The manuscript "Spatial radionuclide deposition data from the 60 km area around the Chernobyl nuclear power plant: results from a sampling survey in 1987" by Kashparov et al. describes the values of various radionuclides from samples obtained in 1987 in the broader region surrounding of the Chernobyl nuclear power plant. The presented dataset is very valuable by itself due to its uniqueness, but nevertheless, the authors present several options for its utilization in the future. The manuscript is well written: I especially appreciate that the authors provide an extensive introduction/background.

[Figure]

In the following review I only state a few comments and technical correction from which the manuscript could benefit.

The landing page of the dataset is well prepared and contains all the relevant information for future users. The dataset itself is well prepared and contains all the data, which is described in the manuscript, except geospatial data (see Specific Comments below). I compliment the authors on the carefully prepared and very clear metadata file. I do have two comment regarding the access to the dataset and the provided data itself, which are posted in the Specific comments section of the review.

Specific comments - Manuscript

Fig. 1 is not very clear. If possible, I suggest the authors modify the original map in a way, that the figure will be readable (enlarge text, indicate all the locations that are mentioned in the manuscript).

L168: Easting and northing data is not included in the dataset! For details see the last Specific comment regarding the dataset.

L168-170: Personally, I think you did a really nice job in creating the Table that is included in the "Spatial_radionuclide_deposition_metadata" document, which accompanies the dataset. I suggest you to include it in this part of the manuscript or in Section 4, as it allows the reader to rapidly understand the meaning of the column headers and the used units (without reading the supporting material). I also suggest to the authors to include in the manuscript a few sentences describing the used data format (e.g. the data is presented in a form of an Excel table etc.).

Specific comments - Dataset

At present (2nd half of February), the data repository requires registration in order to access the dataset and accompanying metadata. As ESSD recommends "two-click" access (see Section 3.1 in https://doi.org/10.5194/essd-10-2275-2018), I suggest the authors consult the Editor, if access in its present state is acceptable.

[Figure]

The dataset in its present state does have one shortcoming, which hinders its use by other users, as it does not contain geospatial data (despite the description at L168 in the manuscript). If the authors will not add northing and easting, they should at least state the coordinates of point zero (ChNPP), so later users can use the provided angles and distance to geolocate the datapoints. If the authors will add northing and easting, a short statement specifying the used coordinate system (possibly by stating the EPSG number) should be added for clarity in the manuscript and in the metadata description.

Technical corrections

L2: remove dot after 1987

L14-15: Replace "Spatial radionuclide deposition data from the 60 km area around the Chernobyl nuclear power plant: results from a sampling survey in 1987" with "Spatial radionuclide deposition data from the 60 km radial area around the Chernobyl nuclear power plant, 1987", as the latter is the name of the dataset provided at https://doi.org/10.5285/a408ac9d-763e-4f4c-ba72-73bc2d1f596d.

L19: Should ". . . include information on sample sites, dose rate . . ." be ". . . include information from sample sites, such as: dose rate . . ."?

L33: "Chernobyl, 1996" should be "Chernobyl Nuclear Power Plant and RBMK reactors, 1996"?

L57-58: I suggest changing ". . . the closest observations were for a distance of more than 100 km away to the west . . ." to ". . . the closest observations were more than 100 km away to the west . . ."

L64: I would omit "fission products" as it is a repetition from L62.

L69: What does the "c." refer to?

L76: Should ". . . including, 40 . . ." be ". . . including 40 . . ."?

L78: ". . . Ukraine. . . ." should be ". . . Ukraine, . . ."

L82: "... Ukraine. ..." should be "... Ukraine, ..."

L102: "... 60-km ..." should be "... 60 km ..."

L117: 14.00-17.00 hours?

L129: I would use "... to identify areas ..." instead of "... to identifying areas ..."

L162: "Figure" should be "Figures"

L163: The authors already describe the acronym UIAR in L18

L177: "Figure" should be "Figures"

L387: The hyperlink includes the ";" symbol and consequentially does not work. Make sure to provide a working link in the revised manuscript.

L389: The hyperlink includes the ";" symbol and consequentially does not work. Make sure to provide a working link in the revised manuscript.

---

## Referee Comment (RC3) · Anonymous Referee #3 · 28 Feb 2020

General Comments

The reviewed manuscript presents the results of radionuclide activity surveys conducted on surficial soils in April and May of 1987 within a 60 km radius of the Chernobyl nuclear power plant, which experienced a catastrophic release of fuel and fission products beginning on April 26, 1986. The stated goal of the authors is to provide the resultant dataset and methodological details specifically to inform dose reconstructions oriented toward human and wildlife impact evaluations and management. Overall the manuscript is well structured and written. The authors presented a detailed overview of

the accident and radionuclide emission timeline, including sufficient information to orient the reader on the fuel emission and remediation, meteorological and depositional processes that contributed to the resultant spatio-temporal pattern of fuel/fission product fallout in the study area. Methodological details were clear, but too brief (a moderate issue), and the connectivity between the dataset and the target applications were well articulated. The data access portal is easy to use, and the dataset and attendant metadata are well organized, but spatial data reporting was insufficient (a moderate issue). Figures were used effectively throughout the manuscript, but in some cases were difficult to read (a minor issue). For these reasons (detailed below) I recommend publication after major revisions.

Specific Comments

The following moderate to minor issues should be addressed in the revised manuscript:

1. Methodological details were insufficient to fully evaluate the gamma spectrometry analyses used to estimate radionuclide activities (moderate revisions). The authors only reported on gamma spectrometer device and sample geometry, however further details on instrument calibration and spectral analysis procedures are necessary to evaluate the approach used to estimate activities and measurement error.

2. Methodological details were insufficient to fully evaluate the accuracy of the sample site locations, and sample site locations were reported in a manner that does not readily support spatial applications of the fallout radionuclide dataset (moderate revisions). The authors used a local, radial coordinate system centered on the Chernobyl plant with 10 degrees spacing between radians and a fixed spacing scheme along radians. Sample location were chosen by superimposing this scheme on 'maps and [the] local landscape,' and reported using only the study's local polar coordinate system. The precision of sample locations generated in this manner is likely quite low. Furthermore, without any additional information, dataset users that convert these local coordinates to values in a geographic coordinate system will each introduce further error. I suggest
that the authors report their study locations using a specified geographic coordinate system, and detail the manner in which this conversion was produced, including an estimate of location error.

Technical Corrections

Many of the figures are difficult to read and/or have minor structural issues Please do not include any text that is unreadable because of size/resolution issues. If text is necessary, then it must be large enough to read (e.g. Figure 1 lat/long, scale, legend labels, etc.). Also, a small panel illustrating the study location in the broader geographic region would be helpful in Figure 1. In Figure 5 please label each axis in the same fashion.

---

## Referee Comment (RC4) · Anonymous Referee #4 · 22 Apr 2020

This accompanying manuscript for the dataset "Spatial radionuclide deposition data from the 60 km area around the Chernobyl nuclear power plant: results from a sampling survey in 1987" by Valery Kashparov et al. is a most valuable study. It is obviously written with profound knowledge of the circumstances of the accidental release of the vast amounts of radioactive material. I am not well informed which other comparable datasets might be publicly available, but it is certainly the first time I see such a detailed assessment and I find it most valuable from various angles- be it the preparation for future accidental releases, radio-ecological assessments, or other special circumstances

like the wildfires currently affecting the region of the nuclear accident. Consequently, I would not want to delay or hinder publication of this dataset, and I have only very few remarks. Obviously, the history of the dataset is a bit complicated (one wonders about the 33 years since analysis), but I got the impression that the origin and ownership of the data has been sufficiently described. Yet, it is still not entirely clear to me what the affiliation of the authors was at the time of sample acquisition, and how responsibilities were distributed. An "author contribution" section, if supported by the journal, might be a good addition. I would like to highlight that I find the development of a "proxy" for alpha-emitting radionuclides a very useful approach for future considerations of radiation safety measures following accidental nuclear fuel release. Detailed comments: 1) The information that I was missing most was a more detailed description of the gamma spectrometry methods. It would be important to know which emission lines were used for which nuclide; which emission probability (if included in the calibration), and which half-lives were used for correction to the release date. These missing pieces are listed in the order of importance. Emission lines are crucial; emission probabilities are optional; and for half-lives, the information is basically there, just not stated explicitly where the correction in mentioned. The more background information there is, the more likely it gets that the dataset can be made comparable with other, similar datasets. If the same emission line, same emission probability and same half-life have been used, one has a much better handle on comparability. One should also consider the aspect that this dataset may become a template for organising similar monitoring programmes in the future, in which case it would be most useful to have the right emission lines at hand. 2) In line 115: A source for these very specific numbers is missing. 3) In line 168: Please remove northing, easting- this is not contained in the dataset I downloaded. The angle and distance are sufficient to reconstruct the location, once a central co-ordinate is given. Northing and easting would be nice to have, but are no reason to delay publication. 4) Figure 8 and 9: I struggle a bit with the interpolation. To me it looks like a large number of measurement points cause a local anomaly, mostly a decrease, in the interpolated values. Why is the algorithm (which algorithm, by the

way) overestimating values over such large areas? Have missing values been actually excluded, or do they go in as zero?

In general, I found the manuscript to be well written and very clear, and I hope to see it published soon.

———————————————

---

## Author Comment (AC1) · 14 May 2020

General comments Anonymous referee 1 This paper describes in detail the spatial radioactive contamination by condensation and fuel of the fallout caused by the Chernobyl accident in 1986, i.e. comparing 144Ce and 137Cs. Making these data available is, as nicely described in chapter 3 "Use of the data" important for assessing the long term effect of radiation exposure of the surrounding landscape including wildlife. The introduction is well written and interesting to read. As a geologist I was missing that today 137Cs deposited in 1986 is commonly used in areas far away from ChNPP to

date sediment layers for environmental reconstructions. JC Thanks for your comments. Whilst we accept that Cs-137 is used for sediment dating we do not think adding a comment to this effect to a paper on data close to the Chernobyl accident is required. I favor Figure 9 because here you can see the development over time (May vs. August 1986), whereas other figures show only the static situation reconstructed for 6th May 1986. When I looked into the data provided, I found a csv table with 20 parameters listed for 491 measurements between 15.05.1987 and 08.06.1987. 49 entries had ID's but no data. Metadata provide explanations and units as well as methods for the shown parameters. In the metadata it was described that missing values are due to water bodies. In the manuscript the authors state that the data include northing and easting, but I could not find coordinates in the data set. Is this missing by mistake? JC No, eastings and northings are not presented. The data are presented as a radial network (i.e. angle and distance from the ChNPP are given). This was a mistake in the text which has been amended. Overall the study is presented in a good way. My concern is that the data are presented as "corrected to 6th May 1986", but obviously based on measurements roughly one year later in 1987. If the data are extrapolations back in time, the authors should describe in detail their methods how they calculated/corrected the values presented in the figures. JC This information has been added at the end of section 2.2. Specific comments: Line 20-21 is this a redundant listing of "caesium-134 and caesium-137" or is there a striking difference? If so, maybe few words explaining why would help. JC This is not redundant and text has been clarified Line 22 You used exactly the same sentences as in the previous paper in ESSD. Please specify "them" in this context. JC Text amended Line 35 Please provide a rough estimate of the vast area size. JC Text amended Line 105-111 Describe how many samples and the spatial resolution of sampling (compare lines 159-160) This paragraph discusses previous studies (not the work reported here) – text amended to hopefully remove any potential confusion. Line 168 I could not find Northing or Easting in the data set. JC Northing or Easting are not in the dataset so these words removed. Line 178 More precise for "regularly" in which temporal resolution? Did sampling take place at exactly the same

locations? The photo shows that the upper column of the soil and grass was sampled. How did the resampling account for accumulation on top of the contaminated layer in subsequent years? JC Clarified that these data are not reported here and are not available Line 198 – 200 – clarification on why data is not available – embargo or not processed? JC As noted in the text these samples were sent to laboratories across the Soviet Union – which is no longer one country (and historically was not an 'open' nation) Line 208 – 210 – uncertainty seems to be high - are there other means to check, whether the uncertainty could be limited? How did you calculate the 50%, is it standard deviation between 5 samples?? JC The text has been amended to describe this more clearly and a reference added to the methodology. Figure 5b. – Why is the R2 = 0.25 not discussed? JC The lower trend for 137Cs with distance was noted in text – but text now amended to acknowledge the R2 value Line 230 – 232, which is associated with the plot, does not include specifics. JC Apologies – but we do not understand the reviewers comment. We have reviewed the text around what were lines 230-232 and cannot identify an issue. Technical comments: Line 33 Is this the correct citation format (Chernobyl, 1996)? JCReference replaced. Line 70 . . ."radiocaesium"? JC Spelling mistake corrected Figure 1. It would help to remove blue color from legend, if it is not used, or use different color instead, because it is too close to the blue of the rivers and lakes. Is there a limit at the top of the legend? JC Figure amended as requested. Table 1. Scientific notation seems not very reader-friendly and the table seems long compared to the intended message. It would help if you could reduce it to a smaller number or highlight entries according to a meaningful criterion. JC Format has been changed. However, information in the table is useful to readers and we have not further amended Line 181-182 repetitive statement to line 162? JC duplicate 489 deleted from line 162.

General comments Anonymous referee 2 General comments The manuscript "Spatial radionuclide deposition data from the 60 km area around the Chernobyl nuclear power plant: results from a sampling survey in 1987" by Kashparov et al. describes the values of various radionuclides from samples obtained in 1987 in the broader region surrounding of the Chernobyl nuclear power plant. The presented dataset is very valuable by itself due to its uniqueness, but nevertheless, the authors present several options for its utilization in the future. The manuscript is well written: I especially appreciate that the authors provide an extensive introduction/background. In the following review I only state a few comments and technical correction from which the manuscript could benefit. The landing page of the dataset is well prepared and contains all the relevant information for future users. The dataset itself is well prepared and contains all the data, which is described in the manuscript, except geospatial data (see Specific Comments below). I compliment the authors on the carefully prepared and very clear metadata file. I do have two comment regarding the access to the dataset and the provided data itself, which are posted in the Specific comments section of the review.

Specific comments – Manuscript Fig. 1 is not very clear. If possible, I suggest the authors modify the original map in a way, that the figure will be readable (enlarge text, indicate all the locations that are mentioned in the manuscript). Figure amended as requested L168: Easting and northing data is not included in the dataset! For details see the last Specific comment regarding the dataset. Please see response to Reviewer 1 L168-170: Personally, I think you did a really nice job in creating the Table that is included in the "Spatial_radionuclide_deposition_metadata" document, which accompanies the dataset. I suggest you to include it in this part of the manuscript or in Section 4, as it allows the reader to rapidly understand the meaning of the column headers and the used units (without reading the supporting material). I also suggest to the authors to include in the manuscript a few sentences describing the used data format (e.g. the data is presented in a form of an Excel table etc.). We have added as supplementary information to the paper Specific comments – Dataset At present (2nd half of February), the data repository requires registration in order to access the dataset and accompanying metadata. As ESSD recommends "two-click" access (see Section 3.1 in https://doi.org/10.5194/essd-10-2275-2018), I suggest the authors consult the Editor, if access in its present state is acceptable. As the editor knows we have previous published in ESSD linking to data on the INSPIRE compliant and Core Trust Seal approved EIDC repository. The dataset in its present state does have one shortcoming, which hinders its use by other users, as it does not contain geospatial data (despite the description at L168 in the manuscript). If the authors will not add northing and easting, they should at least state the coordinates of point zero (ChNPP), so later users can use the provided angles and distance to geolocate the datapoints. If the authors will add northing and easting, a short statement specifying the used coordinate system (possibly by stating the EPSG number) should be added for clarity in the manuscript and in the metadata description. See response above Technical corrections L2: remove dot after 1987 JC Full stop removed L14-15: Replace "Spatial radionuclide deposition data from the 60 km area around the Chernobyl nuclear power plant: results from a sampling survey in 1987" with "Spatial radionuclide deposition data from the 60 km radial area around the Chernobyl nuclear power plant, 1987", as the latter is the name of the dataset provided at https://doi.org/10.5285/a408ac9d-763e-4f4c-ba72-73bc2d1f596d. JC Replaced here and in the title of the manuscript L19: Should ". . . include information on sample sites, dose rate . . ." be ". . . include information from sample sites, such as: dose rate . . ."? JC. No, we mean site information such as unique identifier and location in relation to the ChNPP. I have changed this to 'include sample site information, dose rate,. . .' L33: "Chernobyl, 1996" should be "Chernobyl Nuclear Power Plant and RBMK reactors, 1996"? JC Amended L57-58: I suggest changing ". . . the closest observations were for a distance of more than 100 km away to the west . . ." to ". . . the closest observations were more than 100 km away to the west . . ." JC. 'for a distance' deleted L64: I would omit "fission products" as it is a repetition from L62. JC 'fission products' deleted L69: What does the "c." refer to? JC c means circa (from Latin, meaning 'around, about, roughly, approximately') – frequently abbreviated to c. For clarity I have replaced with approximately. L76: Should ". . . including, 40 . . ." be ". . . including 40 . . ."? JC ',' removed L78: ". . . Ukraine. . . ." should be ". . . Ukraine, . . ." JC Yes, updated L82: ". . . Ukraine. . . ." should be ". . . Ukraine, . . ." JC Yes, updated L102: ". . . 60-km . . ." should be ". . . 60 km . . ." JC Corrected L117: 14.00-17.00 hours? JC 'hours' added for clarity L129: I would use ". .

. to identify areas . . ." instead of ". . . to identifying areas . . ." JC 'ing' deleted L162: "Figure" should be "Figures" JC Changed to (Figures 3 and 4). L163: The authors already describe the acronym UIAR in L18 JC. Agreed - text updated L177: "Figure" should be "Figures" JC Changed to (Figures 3 and 4). L387: The hyperlink includes the ";" symbol and consequentially does not work. Make sure to provide a working link in the revised manuscript. JC. I checked the link - there is no ";" and the link provided opens correctly L389: The hyperlink includes the ";" symbol and consequentially does not work. Make sure to provide a working link in the revised manuscript.' JC. I checked the link - the ";" is not part of the link and the link provided opens correctly

Reviewer 3 General Comments The reviewed manuscript presents the results of radionuclide activity surveys conducted on surficial soils in April and May of 1987 within a 60 km radius of the Chernobyl nuclear power plant, which experienced a catastrophic release of fuel and fission products beginning on April 26, 1986. The stated goal of the authors is to provide the resultant dataset and methodological details specifically to inform dose reconstructions oriented toward human and wildlife impact evaluations and management. Overall the manuscript is well structured and written. The authors presented a detailed overview of the accident and radionuclide emission timeline, including sufficient information to orient the reader on the fuel emission and remediation, meteorological and depositional processes that contributed to the resultant spatio-temporal pattern of fuel/fission product fallout in the study area. Methodological details were clear, but too brief (a moderate issue), and the connectivity between the dataset and the target applications were well articulated. The data access portal is easy to use, and the dataset and attendant metadata are well organized, but spatial data reporting was insufficient (a moderate issue). Figures were used effectively throughout the manuscript, but in some cases were difficult to read (a minor issue). For these reasons (detailed below) I recommend publication after major revisions.

Reviewer 3 Specific Comments The following moderate to minor issues should be addressed in the revised manuscript:

1. Methodological details were insufficient to fully evaluate the gamma spectrometry analyses used to estimate radionuclide activities (moderate revisions). The authors only reported on gamma spectrometer device and sample geometry, however further details on instrument calibration and spectral analysis procedures are necessary to evaluate the approach used to estimate activities and measurement error. Text added

2. Sample location were chosen by superimposing this scheme on 'maps and [the] local landscape,' and reported using only the study's local polar coordinate system. The precision of sample locations generated in this manner is likely quite low. Furthermore, without any additional information, dataset users that convert these local coordinates to values in a geographic coordinate system will each introduce further error. I suggest that the authors report their study locations using a specified geographic coordinate system, and detail the manner in which this conversion was produced, including an estimate of location error. Information on precision of sample location was given.

Technical Corrections Many of the figures are difficult to read and/or have minor structural issues Please do not include any text that is unreadable because of size/resolution issues. If text is necessary, then it must be large enough to read (e.g. Figure 1 lat/long, scale, legend labels, etc.). Also, a small panel illustrating the study location in the broader geographic region would be helpful in Figure 1. In Figure 5 please label each axis in the same fashion. Figures amended.

Author response to Anonymous Referee #4 JC We thank the anonymous referee for their positive feedback and constructive suggestions.

Referee comment: Yet, it is still not entirely clear to me what the affiliation of the authors was at the time of sample acquisition, and how responsibilities were distributed. An "author contribution" section, if supported by the journal, might be a good addition.

Author comment: author contribution section added below 4. Data availability section.

Author contribution. Soil samples were collected by the USSR Ministry of Defence and

delivered to UIAR. Sample preparation, analysis and data interpretation was carried out by UIAR staff contributing as follows: Kashparov, Levchuk, Protsak, - sample preparation, measurement of radionuclide activity concentrations in samples; Kashparov - analysis of results; Zhurba - database creation and preparation of the manuscript figures (maps). The manuscript was prepared by Chaplow, Beresford, Kashparov, Levchuk and Zhurba.

Specific comments:

Referee comment 1) The information that I was missing most was a more detailed description of the gamma spectrometry methods. It would be important to know which emission lines were used for which nuclide; which emission probability (if included in the calibration), and which half-lives were used for correction to the release date. These missing pieces are listed in the order of importance. Emission lines are crucial; emission probabilities are optional; and for half-lives, the information is basically there, just not stated explicitly where the correction in mentioned. The more background information there is, the more likely it gets that the dataset can be made comparable with other, similar datasets. If the same emission line, same emission probability and same half-life have been used, one has a much better handle on comparability. One should also consider the aspect that this dataset may become a template for organising similar monitoring programmes in the future, in which case it would be most useful to have the right emission lines at hand.

Author comment: the manuscript has been amended to include further information on the gamma spectrometry methods - both in the methods text and also with the addition of extra information as Appendix 2/

2.2 Analysis

Using a high-purity germanium detector (GEM-30185, ORTEC, USA) and a multichannel analyser "ADCAM-300" (ORTEC, USA), the activity concentration of gamma emitting radionuclides (zirconium-95 (95Zr), niobium-95 (95Nb), ruthenium-106 (106Ru),

caesium-134 (134Cs), caesium-137, (137Cs) cerium-144 (144Ce)) was determined in one soil sample from each sampling site. Information on gamma lines used in the analyses and radioisotope half-lives assumed for decay correction are presented in Appendix 2. Soil samples were analysed in a 1 litre Marinelli container. The other four cores were sent to different laboratories in the Soviet Union (data for these cores are unfortunately not available). Using a 1M NH4Ac solution (pH 7) a 100 g subsample of soil was leached (solid: liquid ratio 1:5). The resultant leachate solution was shaken for 1 hour and then left at room temperature for 1 day before filtering through ashless filter paper (3-5 $\mu$m). The filtrate was then put into a suitable container for gamma analysis to determine the fraction of exchangeable 134,137Cs. Measured activity concentrations were reported at 68% confidence level (which equates to one standard deviation).

Decay radiation information from the master library, integrated in spectrum analysing software tool Gelicam (EG&G ORTEC, USA), was used in gamma-analyses. Activities of 106Ru and 137Cs in samples were estimated via their gamma radiation emitting progenies 106Rh and 137mBa, respectively.

Calibration of the spectrometer was conducted using certified standards (soil equivalent multi-radionuclide standard, V. G. Khlopin Radium Institute, Russia). Quality assurance/quality control procedures included regular monitoring of the system performance, efficiency, background and full width at half maximum (FWHM) for the 144Ce, 137Cs and 95Nb photo peaks. To validate accuracy and precision of the method employed for 137Cs activity concentration measurements, quality control samples (i.e., different matrix samples including water, soil and sawdust spiked with known certified activities of radionuclides) and Certified Reference Materials (CRM) were analysed alongside the samples. Analysis of IAEA CRMs showed satisfactory results for radionuclide mean activity concentrations with results being within the 95% confidence interval; the limit of detection for 137Cs in all samples was 1 Bq. Subsamples were analysed in a different laboratory (USSR Ministry of Defence) and results for the two laboratories were within the error of determination.

Appendix 2. Decay radiation information from the master library, integrated in spectrum analysing software tool Gelicam (EG&G ORTEC, USA), used in gamma-analyses. Activities of 106Ru and 137Cs in samples were estimated via their gamma radiation emitting progenies 106Rh and 137mBa, respectively (see attached zip supplement for table)

Referee comment 2) In line 115: A source for these very specific numbers is missing.

Author comment: Reference added to manuscript as Aleksakhin et al., 2001 and added to References section.

In the initial phase after the accident (before 7th May 1986) 99195 people were evacuated from 113 settlements including 11358 people from 51 villages in Belarus and 87 837 people from 62 settlements in Ukraine (including about 45 thousand people evacuated between 14.00-17.00 hours on April 27 from the town of Pripyat located 4 km from the ChNPP) (Aleksakhin et al., 2001).

References

Aleksakhin R.M., Buldakov L.A., Gubanov V.A., Drozhko E.G., Ilyin L.A., Kryshev I.I., Linge I.I., Romanov G.N., Savkin M .N., Saurov M.M., Tikhomirov F.A., Kholina Yu.B. 2001. Major radiation accidents: consequences and protective measures. Edited by L.A. Ilyin and V.A. Gubina. book published in Moscow, Publishing House IzdAT. 752 p. (data from p. 481). ISBN 5-86656-113-1 http://elib.biblioatom.ru/text/krupnye-radiatsionnye-avarii_2001/go,0/

Referee comment 3) In line 168: Please remove northing, easting- this is not contained in the dataset I downloaded. The angle and distance are sufficient to reconstruct the location, once a central co-ordinate is given. Northing and easting would be nice to have, but are no reason to delay publication.

Author comment: Apologies, this was a mistake; eastings and northings are not available as noted by all reviewers and this has been removed.

Referee comment 4) Figure 8 and 9: I struggle a bit with the interpolation. To me it looks like a large number of measurement points cause a local anomaly, mostly a decrease, in the interpolated values. Why is the algorithm (which algorithm, by the way) overestimating values over such large areas? Have missing values been actually excluded, or do they go in as zero?

Author comment: The reviewer's comment was not totally clear to us as they seem to contradict 'mostly a decrease' and then 'overestimation'. However, we have reviewed the text and added some information about the interpolation method and also a short clarifier in the figure legends with regard to the white area in the centre of the interpolate surface. For sites from which no samples were collected (e.g. waterbodies) nothing was included in the interpolation (I.e. no assumed value of zero was used as the reviewer questions).

As an example of the application of the data in this manner, Figure 8 presents the estimated deposition of 238Pu; Figure 8 was prepared using the TIN (triangulated ir-regular network) interpolation within MAPINFO. The first maps of 90Sr and 239+240Pu surface contamination from the Chernobyl accident were prepared in the frame of an international project (IAEA, 1992) in a similar way.

Figure 8. The fallout density of 238Pu (kBq m-2) corrected to 6th May 1986; estimated from measurements of 144Ce in soil and estimated activity concentrations in the fuel of the ChNNP reactor number four (note no data were available for less than 5 km from ChNPP and no interpolation for this area has been attempted).

Figure 9. Spatial distribution, interpolated as for Figure 8, of effective dose rate within the 60 km zone around the ChNPP on 10th May 1986 (a) and 10th August 1986 (b). Note no data were available for less than 5 km from ChNPP and no interpolation for this area has been attempted.

Please also note the supplement to this comment:

https://www.earth-syst-sci-data-discuss.net/essd-2019-174/essd-2019-174-AC1-supplement.zip